# Evaluation of Biocompatible Materials for Enhanced Mesenchymal Stem Cell Expansion: Collagen-Coated Alginate Microcarriers and PLGA Nanofibers

**DOI:** 10.3390/biom15030345

**Published:** 2025-02-27

**Authors:** Manuel Jaime-Rodríguez, María Luisa Del Prado-Audelo, Norma Angélica Sosa-Hernández, Dulce Patricia Anaya-Trejo, Luis Jesús Villarreal-Gómez, Ángel Humberto Cabrera-Ramírez, Jesus Augusto Ruiz-Aguirre, Israel Núñez-Tapia, Marek Puskar, Emily Marques dos Reis, Silvia Letasiova, Rocío Alejandra Chávez-Santoscoy

**Affiliations:** 1Tecnológico de Monterrey, School of Engineering and Science, Av. Eugenio Garza Sada 2501 Sur, Monterrey 64849, Nuevo León, Mexico; mjaimer@tec.mx (M.J.-R.);; 2Biomedical Sciences Department, Universidad Nacional Autónoma de México, Av. Universidad 3004, Coyoacán, Ciudad de Mexico 04510, Mexico; 3Engineering and Technology Science Faculty, Universidad Autónoma de Baja California, Calzada Universidad 14418, Parque Industrial, Tijuana 22424, Baja California, Mexico; 4Centro de Investigación y Asistencia en Tecnología y Diseño del Estado de Jalisco, A.C. Parque Científico Tecnológico de Yucatán, Km.5 Carretera, Sierra Papacal-Chuburná, Chuburná, Mérida 97302, Yucatán, Mexico; 5Materials Research Institute, Universidad Nacional Autónoma de México, Circuito Exterior S/N, Circuito de la Investigación Científica, Coyoacán, Ciudad de Mexico 04510, Mexico; 6MatTek Europe, Mlynske Nivy 73, 82105 Bratislava, Slovakia

**Keywords:** mesenchymal stem cells, adherence, alginate, PLGA, biocompatibility, regenerative medicine, cultivated meat

## Abstract

Mesenchymal stem cells (MSCs) hold significant potential in regenerative medicine, tissue engineering, and cultivated meat production. However, large-scale MSC production is limited by their need for surface adherence during growth. This study evaluates two biocompatible materials—collagen-coated alginate microcarriers and polylactic-co-glycolic acid (PLGA) nanofibers—as novel growth substrates to enhance MSC proliferation. Physicochemical characterization confirmed successful collagen integration on both materials. In vitro, bone marrow-derived MSCs (bmMSCs) cultured on collagen-coated alginate microcarriers exhibited significantly enhanced growth compared to commercial microcarriers, while PLGA nanofibers supported bmMSC growth comparable to traditional growth surfaces. Scanning Electron Microscopy (SEM) revealed that bmMSCs adhered not only to the surface but also grew within the porous structure of the alginate microcarriers. Mycoplasma testing confirmed that the bmMSCs were free from contamination. Both materials were assessed for biocompatibility using ISO-10993 guidelines, demonstrating no skin or ocular irritation, supporting their potential for in situ applications in clinical and therapeutic settings. This study highlights the promise of collagen-coated alginate microcarriers and PLGA nanofibers for scalable MSC production, offering efficient, biocompatible alternatives to traditional growth surfaces in regenerative medicine and cultivated meat manufacturing. Future research should focus on optimizing these materials for larger-scale production and exploring specific applications in therapeutic and food sectors.

## 1. Introduction

Mesenchymal Stem Cells (MSCs) are non-hematopoietic stem cells that possess significant proliferative potential and the ability to differentiate into various cell types, including adipocytes, cardiomyocytes, chondrocytes, myocytes, and neurons. Due to their regenerative potential, MSCs have been proposed as blocks for therapeutic applications, such as bone regeneration, neuron and cartilage repair, liver and cardiovascular disease treatment, and renewal of meniscus, tendons, and ligaments [1]. Additionally, MSCs modulate inflammatory responses by secreting a wide range of cytokines, chemokines, growth factors, and proteomes.

Mesenchymal stem cells (MSCs) are multipotent stromal cells derived from various sources, including bone marrow, adipose tissue, and umbilical cord, each exhibiting distinct biological properties and therapeutic potential. The tissue of origin significantly influences the immunomodulatory capacity, differentiation potential, and regenerative efficacy of MSCs. For instance, bone marrow-derived MSCs (bmMSCs) have been widely studied for their role in hematopoietic support and osteogenic differentiation, while adipose-derived MSCs (AD-MSCs) demonstrate superior proliferative capacity and angiogenic potential, making them promising for soft tissue regeneration. Meanwhile, umbilical cord-derived MSCs (UC-MSCs) are characterized by lower immunogenicity and enhanced paracrine activity, contributing to their potential in immunotherapy and neuroprotection [2,3]. Therefore, identifying MSC sources with higher proliferative efficiency is essential for improving therapeutic scalability.

In vitro, in vivo, and clinical evidence supports the use of MSCs as an adequate treatment for immune disorders, including allograft rejection, type 1 and 2 diabetes, acute graft versus host disease, peritonitis, endotoxemia, ischemia, acute liver injury, arthritis, atherosclerosis, Crohn’s disease, and multiple sclerosis [4]. Moreover, MSCs have been shown to downregulate the inflammatory response mediated by lymphocytes in in vitro models [5]. These findings were corroborated subsequently in mouse models and phase I and II clinical trials for the mentioned applications [6].

In addition to their therapeutic applications and potential to generate muscle and adipose tissue, MSCs have been proposed as building blocks for producing cultured meat for human consumption. Cultured meat is based on muscle generated de novo from MSCs under laboratory conditions [7]. This process eliminates animal slaughter and reduces the environmental impact of traditional meat production, while maintaining protein content and composition comparable to native meat. However, a fundamental characteristic that has limited the scale-up process required for producing sufficient MSCs for their therapeutic, immunomodulatory, and cultivated meat applications is the fact that MSCs require adherence to a surface to grow in laboratory conditions [8]. This adherence is a disadvantage for the use of MSCs as all the mentioned uses require more than 1 × 10^9^ cells per application batch. These cell quantities cannot be efficiently produced with multi-tray systems or roller bottles [9].

The main limitation is, therefore, the need to optimize the growth surface area about the container volume. To overcome this scale-up problem, it is common to use microcarriers, microscopic spheres that provide an adhesive surface and are small enough to be suspended in cell culture media under agitation. The implementation of these structures can efficiently optimize the surface-to-volume ratio [10]. However, these strategies generally consider materials such as borosilicates, glass, etc., which ultimately require an additional unit of operation to detach the cells for use in the applications. This problem can be significantly reduced by implementing biocompatible scaffolds such as alginate, which has been extensively used as a scaffold material for producing different types of cells due to its well-documented properties, including coating stability, cellular adhesion, biocompatibility, and biodegradability. In this article alginate microcarriers are tested in conditions used to culture adherent cells in stirred bioreactors.

Additionally, differentiation into specialized cells is critical for various applications, but achieving consistent results has been challenging. In response, nanostructured surfaces have been developed to improve the control and efficiency of the differentiation process by simulating the microenvironment of target tissues, with electrospun PLGA nanofibers emerging as a promising option [11]. Electrospinning enables the continuous production of polymer fibers with nanoscale diameters, driven by a high-voltage electric field and controlled polymer flow rates. PLGA nanofibers have demonstrated remarkable biocompatibility and exceptional cell attachment capabilities, making them suitable growth surfaces for MSCs [12].

On one hand, both alginate microsphere and PLGA nanofiber production promise to address key challenges in mesenchymal stem cell (MSC) applications, from therapeutic interventions to cultured meat production. Culturing MSCs in biocompatible and biodegradable materials facilitates in situ cellular applications and streamlines processes compared to traditional methods. By avoiding trypsinization and other complex downstream operations, a significant cost reduction of 8–13% can be achieved, while maintaining cell viability and surface receptor integrity [13]. On the other hand, electrospray and electrospinning are promising techniques that are known for their stability, scalability in batch production, reproducibility, minimization of swelling effects, and a superior efficiency in the generation of microscopic particles. These are robust materials to support and enhance MSC cultivation.

The objectives of this study are: 1. To evaluate the efficacy of electrosprayed alginate microcarriers and electrospun PLGA nanofibers, along with their collagen-coated iterations, as highly efficient proliferation surfaces for bone marrow-derived MSCs (bmMSCs), 2. To physicochemically characterize the developed biomaterials. 3. To compare cell proliferation efficiency with existing commercial alternatives. 4. To test the biomaterials’ biocompatibility following ISO-10993 guidelines [14]. Additionally, we discuss the scalability potential of these methods for direct in situ applications in the future. Furthermore, our bmMSC culture protocol avoids antibiotics to mitigate potential issues related to drug resistance generation and immunosuppression in the end user. Most importantly, this study was conducted without fetal bovine serum (FBS) to address reproducibility issues, offer competitive pricing, and increase the purity of the final product.

## 2. Materials and Methods

### 2.1. Materials

In this study, bone marrow-derived Mesenchymal stem cells (bmMSCs) were isolated from 6-week-old American Yorkshire pig bone marrow, and their isolation procedure is detailed below. Sodium alginate (Mw: 12–40 KDa), PLGA:Resomer(R) RG858 S, Poly (D,L-lactide-co-glycolide), ester terminated, lactide/glycolide 81:15 (Mw: 190–240 KDa), Calcium Chloride, Thiazolyl Blue Tetrazolium Bromide (MTT), Ficoll-Paque, chloroform, dimethylformamide, glutaraldehyde, ethanol, methanol, acetic acid, sesame oil, sodium dodecyl sulfate, methyl acetate, hematoxylin-eosin staining and Type I collagen were purchased from Merck Ltd. (Darmstadt, Germany). Meanwhile, cell culture slides, Hoechst 34580, 0.05% recombinant trypsin, 0.02% EDTA solution, HEPES 1M, and PBS pH 7.4 were acquired from Thermo Fisher Scientific Inc. (Waltham, MA, USA). NutriStem XF Media. SoloHill Collagen-Coated Microcarriers and Trypsin Soybean Inhibitor were obtained from Sartorius AG (Göttingen, Germany). Finally, Stemflow Human MSC Analysis Kit, hMSC Positive Cocktail, hMSC Negative Cocktail, and Pharmingen Stain Buffer were purchased from Bencton Dickinson Inc. (Franklin Lake, NJ, USA). EpiDerm and EpiOcular tissues were bought from Mattek In Vitro Life Science Laboratories (Bratislava, Slovakia), S.R.O. 12-well Cell Culture plates were obtained from Biologix Group Ltd. (Shandong, China).

### 2.2. Alginate Microcarriers Production

Alginate microcarriers were obtained by electrospraying following the protocol reported by Xu et al. [15]. Briefly, a 1.25% (*w*/*v*) alginate solution was pumped at a flow rate of 0.35 mL/h using a syringe with 0.84 mm diameter needle attached to a pump (Infusion Syringe Pump IPS series, Inovenso, Cambridge, MA, USA). Electrospraying was executed at 9.5 KV using a power source (MSK-ESPS-30–24V High Voltage, MTI Corporation, Richmond, CA, USA) and an 8 cm distance between the needle tip and the collector solution (2.5% Calcium Chloride). The collector solution was stirred constantly at 350 rpm, maintaining a temperature of 25 °C and a relative humidity of 40%.

### 2.3. Polylactic-Co-Glycolic Acid (PLGA) Nanofibers Production

Electrospinning was to obtain the PLGA nanofibers, following the protocol reported by Yu et al. [16]. Briefly, a 5% (*w*/*v*) PLGA solution was prepared in a binary-solvent system of chloroform and dimethylformamide (80:20). The solution was stirred at 40 °C until complete dissolution, confirmed by the formation of a transparent fluid. The 5% PLGA solution was then pumped at a flow rate of 0.5 mL/h using a system consisting of a 20-gauge syringe needle attached to a pump (Infusion Syringe Pump, IPS series, Inovenso, Cambridge, MA, USA). Electrospinning was carried out at 20 KV using a power source (MSK-ESPS-30–24V High Voltage, MTI Corporation, Richmond, CA, USA) with a 20 cm distance from needle tip to collector, maintaining a temperature of 25 °C and a relative humidity of 40%. A total spinning time of 3 h was employed to produce 10 × 10 cm non-woven structures.

### 2.4. Collagen Coating and Disinfection Procedure

Before the procedure, PLGA nanofibers were cut into the shape of a 7.5 cm^2^ circle (equivalent to a 12 well plate surface), and alginate microcarriers were adjusted to have a total surface of 7.5 cm^2^ for each treatment. The surfaces of PLGA nanofibers and alginate microcarriers were collagen-coated and disinfected before cell application with some modification of the protocol of Hong et al. [17]. Nanofibers and microcarriers were incubated in a 1% collagen aqueous solution for 2 h with moderate agitation. Subsequently, the two surfaces were transferred to a 1% glutaraldehyde solution for 30 min. The pair was washed thrice with Phosphate Buffer Solution (PBS) pH 7.4 and disinfected with a 30-s incubation in a 70% ethanol solution, followed by three washes with PBS pH 7.4. Positive controls: SoloHill collagen-coated microcarriers (Sartorius AG, Göttingen, Germany) were used after disinfection protocol with 70% ethanol; a 12-well cell culture plate without any collagen coating or disinfection protocol.

Collagen-free PLGA nanofibers and alginate microcarriers were produced, adjusted to 7.5 cm^2^, and disinfected with 70% ethanol protocol. Every experimental and control biomaterial underwent a sterility test with 1 mL of MSC NutriStem XF and was incubated at 37 °C for 48 h.

### 2.5. Alginate Microcarriers and PLGA Nanofiber Physicochemical and Structural Characterization

Thermal stability was evaluated using a Thermogravimeter Analyzer (Hi-Res TGA 2950, TA Instruments, New Castle, DE, USA), employing 5 mg of dry samples (collagen, collagen-free alginate microcarriers, collagen-coated alginate microcarriers, collagen-free PLGA nanofiber, and collagen-coated PLGA nanofiber). Tests were conducted from 25 °C to 500 °C at a rate of 10 °C/min under a nitrogen atmosphere. For all the materials evaluated, 50% mass loss was graphically calculated.

On the other hand, the infrared spectra were collected using a Thermo Scientific spectrometer (Nicolet iS10 FT-IR, Waltham, MA, USA). IR spectra of dry collagen-free and collagen-coated alginate microcarriers were collected from 4000 to 500 cm^−1^. Finally, microscopic images of the experimental and control biomaterials, both with and without cells, were obtained using the LAS-1000 inverted microscope (Leica, Germany).

### 2.6. Antibiotic and Serum-Free Mesenchymal Stem Cell Primary Culture and Subculture

Bone marrow was collected from a 6-week-old male American Yorkshire pig by pectoral puncture following the administration of thiopental at a dose of 5 mg/Kg. bmMSCs were isolated using a Ficoll-Paque gradient. Briefly, the bone marrow was diluted in a 1:1 proportion with PBS at pH 7.4 and transferred to Ficoll-Paque. The gradient was centrifuged at 400 g for 40 min at 20 °C. The buffy coat was cultured with MSC NutriStem XF Media at 37 °C with 5% CO_2_. When the cells reached 70–80% confluency, they were trypsinized with 0.05% recombinant trypsin and 0.02% EDTA and incubated at 37 °C for 5 min. The trypsin was inactivated with Trypsin Soybean Inhibitor, and the cells were harvested and expanded in larger flasks for a maximum of 20 passages.

### 2.7. Flow Cytometry Porcine bmMSC Characterization

Porcine bmMSCs were characterized by using the Stemflow Human MSC Analysis Kit (Bencton Dickinson Inc., Franklin Lake, NJ, USA), with the hMSC Positive Cocktail and the hMSC Negative Cocktail (Bencton Dickinson Inc., Franklin Lake, NJ, USA), following kit indications. The stained cell suspension was washed twice with BD Pharmingen Stain Buffer and analyzed on a FACS Aria flow cytometer (Bencton Dickinson, New York, NY, USA).

### 2.8. Mycoplasma Testing

bmMSC cultures were evaluated for Mycoplasma contamination by using indirect Hoechst staining methodology as reported by Young et al. [18]. Briefly, bmMSCs were cultured overnight at 37 °C and 5% CO_2_ in cell culture slides. Cells were fixed with methanol/acetic acid 1:1 and 1:3 solution and stained with Hoechst 33342 (Thermo Fisher Scientific Inc., Waltham, MA, USA). Samples were evaluated with epifluorescence microscope with 350 nm excitation wavelength and 460 nm filter. This same protocol was applied for Mycoplasma-contaminated cells in isolated conditions from bmMSCs.

### 2.9. Alginate Microcarriers and PLGA Nanofiber Growth Kinetics for Biological Characterization

A kinetic was performed thrice with triplicates of 0, 12, 24, 36, 48, 60, 72, 84, 96, 120, 144, 192, 240, 288, 336, 384, 432, 480, and 528-h incubations used for each treatment. The bmMSC suspension was adjusted to a cell density of 2 × 10^5^ cells/mL with Sartorius MSC NutriStem XF Media with 10 mM HEPES and added to each evaluated biomaterial adjusted to 7.5 cm^2^ and incubated at 37 °C.

On one hand, collagen-coated PLGA nanofibers, collagen-free PLGA nanofibers, and the 12-well cell culture plate were directly incubated at 37 °C without agitation during the kinetic incubation time. On the other hand, collagen-coated alginate microcarriers, collagen-free alginate microcarriers, and SoloHill collagen-coated microcarriers (Sartorius AG, Göttingen, Germany) were incubated for 10 cycles of 5 min incubation at 37 °C with agitation (250 rpm) and 1 min incubation at 37 °C; after that, the microcarriers were incubated at 37 °C and 250 rpm agitation for the kinetic incubation times.

After the incubation times, 100 μL of a 200 μg/mL MTT solution was added to each treatment and incubated at 37 °C for 2 h. The produced formazan was diluted with isopropanol, and the absorbance was read at 540 nm wavelength.

### 2.10. Cell Number-Absorbance Standard Curve

To convert the 540 nm absorbance values into bmMSC number, a standard curve of cell number versus absorbance was constructed. bmMSC suspension was adjusted to a cell density of 1.2 × 10^6^ cells/mL with MSC NutriStem XF Media with 10 mM HEPES. Dilutions were performed with medium to obtain the following cell concentrations: 1.2 × 10^6^; 1.0 × 10^6^; 0.8 × 10^6^; 0.6 × 10^6^; 0.4 × 10^6^; 0.2 × 10^6^; 0 × 10^6^ cells/mL. One milliliter of each cell concentration was seeded in triplicate in a 12-well cell culture plate and was incubated at 37 °C for 1 h. After the incubation, MTT solution was added and treated as mentioned in the above section. Standard curve equation to relate cell number to 540 nm absorbance was determined by linear regression using Excel Version 2501.

Additionally, the relative growth efficiency (%) for each treatment was determined for every 96 h incubation by using the total cells from the 12-well cell culture plate after 96 h as reference, and Equation (1) was applied.(1)$$Relative\;Growth\;Efficiency\;(\%)=\frac{96\;h\;average\;total\;cells\;for\;each\;treatment}{96\;h\;average\;total\;cells\;in\;12-well\;Biologix\;Cell\;Culture\;plate}\times 100$$

### 2.11. Alginate Microcarriers and PLGA Nanofiber Scanning Electron Microscopy (SEM)

The bmMSC suspension was adjusted to a cell density of 2 × 10^5^ cells/mL with MSC NutriStem XF Media with 10 mM HEPES. 1 mL of the cell suspension was applied to analyzed biomaterials following the incubation indications mentioned in Section 2.9. After the incubation, the cell culture media was removed, and surfaces were resuspended in a 1% glutaraldehyde solution for 30 min. The surfaces were washed thrice with PBS solution at pH 7.4 and dried for 1 week.

Finally, the cut collagen-coated alginate microcarriers for SEM were obtained by slicing the collagen-coated microcarriers after the culture with cells, and before removing the cell culture media, the surfaces were cut with the help of sterile scalpel blades. The spheres were resuspended in a 1% glutaraldehyde solution for 30 min. The surfaces were washed thrice with PBS solution pH 7.4 and dried for 1 week.

All the samples were coated with a 5 nm gold coat and observed using a MERLIN Scanning Electron Microscope (Zeiss, Cambridge, UK).

### 2.12. Biocompatibility Test

Following ISO 10993 guidelines [14], aqueous and organic extract were obtained from collagen-coated PLGA nanofibers by preparing a 6 cm^2^/mL sesame oil and 0.9% saline solution correspondingly and incubating them at 37 °C for 72 h.

Skin irritation for both material extracts and directly applied materials (25 mg collagen coated-alginate microcarriers and 8mm circle of PLGA nanofiber) was evaluated as reported by Kandarova et al. [19]; briefly, a tridimensional, multilayered, highly differentiated reconstructed human epidermal model (EpiDerm^TM^, MatTek In Vitro Life Science Laboratories, Bratislava, Slovakia) was exposed by triplicate to 100 mL of organic and aqueous extracts from both materials, negative control (PBS), vehicle controls (saline and sesame oil), and positive control (1% SDS prepared in saline, sesame oil and PBS) for 18 h ± 30 min at 37 °C and 5% CO_2_. Afterward, the incubation tissues were rinsed with PBS and incubated for 3 h in 500 mg/mL MTT solution at 37 °C and 5% CO_2_. Formazan was extracted by submerging all the tissues in 2 mL of isopropanol. The resultant absorbance of each tissue was determined at 570 nm and relative viability was calculated as follows:(2)$$Relative\;Viability\;(\%)=\frac{Absorbance\;at\;570\;nm\;for\;each\;treatment}{Mean\;absorbance\;at\;570\;nm\;of\;Negative\;Control\;(PBS)}\times 100$$

Ocular irritation for both material extracts was evaluated as reported by Pobis et al. [20]; briefly, a non-keratinized cornea epithelium with progressively stratified cells from normal human keratinocytes (EpiOcular^TM,^ MatTek In Vitro Life Science Laboratories, Bratislava, Slovakia) was exposed by triplicate to 50 mL of organic and aqueous extracts from both materials, negative control (PBS), vehicle controls (saline and sesame oil), and positive control (methyl acetate and 1% SDS prepared in saline or sesame oil) for 30 min at 37 °C and 5% CO_2_. Tissues were rinsed with PBS and were incubated for 2 h at the same rate. Afterward, the incubation tissues were rinsed with PBS and incubated for 3 h in 500 mg/mL MTT solution at 37 °C and 5% CO_2_. Formazan was extracted by submerging all the tissues in 2 mL of isopropanol. The resultant absorbance of each tissue was determined at 570 nm and relative viability was calculated as mentioned in Formula (2).

Samples for histological staining were washed with PBS, fixed in 4% paraformaldehyde, and embedded in paraffin, and 4 mm were prepared for hematoxylin and eosin tinction.

### 2.13. Statistical Analysis

Data are expressed as the mean ± standard deviation of three independent experiments. Using the statistical software Minitab 19, a One-way ANOVA with means comparison by the Tukey test was performed at a 95% confidence level (α = 0.05).

## 3. Results

### 3.1. Alginate Microcarriers and Polylactic-Co-Glycolic Acid (PLGA) Nanofibers Production

The findings of this study unveil critical insights into the comparative performance of alginate microcarriers and PLGA nanofibers as substrates for bone marrow-derived Mesenchymal Stem Cell (bmMSC) proliferation. Figure 1 illustrates the morphology at different scales of the alginate microcarriers and PLGA nanofibers. On a macro scale, the alginate microcarriers exhibited a granular morphology (Figure 1A), while the PLGA nanofibers comprised laminated structures (Figure 1B). At a micro scale, the alginate microcarriers displayed a well-defined and regular circular shape, with an apparently uniform size (Figure 1C). However, due to the optical characteristics of the PLGA nanofibers (Figure 1D), no further detail of their microscopic structure can be appreciated; therefore, the evaluation of cell viability and adhesion was conducted only at the end of the incubation times using MTT assay and SEM.

### 3.2. Disinfected and Collagen-Coated Alginate Microcarriers and PLGA Nanofibers Production

Having assessed the morphology and ensured the safety of the biomaterials, the efficiency of collagen coating and the suitability to grow adherent cells for alginate microcarriers were evaluated. For this purpose, an incubation of bmMSCs with the biomaterials, using the conditions described in the growth kinetics section, was carried out. The cell adherence in alginate microcarriers was confirmed by microscopic observation (Figure 1E), where bmMSCs are attached to the microsphere surface. On the other hand, Solohill collagen-coated microcarriers (Figure 1F) were used as a reference for the procedure.

Microscopic evaluation of the PLGA nanofibers was hindered by the opacity of the material, as shown in Figure 1.

### 3.3. Alginate Microcarriers and PLGA Nanofibers Physicochemical and Structural Characterization

The integration of collagen coating has a significant effect on the thermogravimetric profile of alginate microcarriers and PLGA nanofibers. Figure 2A,B show the mass-loss behavior versus temperature for the different systems evaluated. Overall, collagen appears to significantly impact the thermal stability of alginate microcarriers and PLGA nanofibers, depending on the specific configuration. In alginate microcarriers, coating with collagen seems to strengthen the structure and thermal stability of the material, increasing from 365 to 498 °C before reaching a 50% loss of its mass. Conversely, in PLGA nanofibers, a negative effect was observed after collagen incorporation, decreasing from 400 to 300 °C.

TGA curves were similar among treatments, but the temperature at which 50% mass loss was reached varied as a function of the type of biomaterial and the addition of collagen: collagen (330 °C), collagen-free alginate microcarriers (365 °C), collagen-coated alginate microcarriers (498 °C), collagen-free PLGA nanofibers (400 °C), and collagen-coated PLGA nanofibers (300 °C).

### 3.4. Antibiotic and Serum-Free Mesenchymal Stem Cell Primary Culture and Flow Cytometry Characterization

Bone marrow was obtained from pigs without causing any harm or death to the donor, and the bmMSCs were isolated and adapted to grow in laboratory conditions without the use of Fetal Bovine Serum or antibiotics during the process. In this research, bmMSC culture was implemented with the use of serum-free media (MSC NutriStem XF Media).

The isolated bmMSCs showed the fibroblastoid morphology typical to these types of cells (Figure 3B) and had the characteristic MSC surface marker profile, as supported by flow cytometry characterization.

The cultured bmMSCs presented the next immunophenotype: CD73 (89.05%), CD90 (93.11%), CD105 (91.92%), CD34 (0.06%), CD45 (0.06%), and HLA-DR (0.06%).

### 3.5. Alginate Microcarriers and PLGA Nanofiber Growth Kinetics

A standard curve of cells versus absorbance was generated to interpret the MTT absorbance as a biological variable (cell number). The resulting standard curve exhibited a determination coefficient (R2) of 0.9907 and a line equation of $y=1\times 10^{6}x+0.0121$ (Appendix A Figure A1). Using this lineal equation, the total cell production was determined for each replicate at different incubation times for each treatment. For a more effective comparison, microcarriers (collagen-coated alginate microcarriers, collagen-free alginate microcarriers, and SoloHill collagen-coated microcarriers) were graphed independently (Figure 4) to the cell surfaces (collagen-coated PLGA nanofibers, collagen-free PLGA nanofibers, and 12-well cell culture).

For the evaluation of statistical significance between the treatments and the different incubation times, Tukey test grouping was performed (Appendix A Table A1 and Table A2). As indicated in Figure 4 and Appendix A Table A2, there is no significant difference in growth between collagen-coated PLGA nanofibers and 12-well cell culture at any incubation time. It can be concluded that the bmMSCs grow at the same rate and efficiency on both the nanofiber and cell culture plastic plates. These kinetic results support bmMSC growth when adhered to PLGA nanofibers and were confirmed by the SEM results. 

Finally, for the evaluation of the total 96 h incubation for all the treatments, Tukey test grouping for the final kinetic results of all treatments was performed (Table 1), and relative growth efficiency was calculated relative to 12-well cell culture surface (Table 2).

In Figure 4 and Appendix A Table A1 bmMSCs grown in alginate microcarriers exhibit greater growth efficiency than commercial Solohill collagen-coated microcarriers cultivated in the same conditions. The 22-day comparative growth kinetic is shown in Figure 5.

### 3.6. Morphology of Alginate Microcarriers and PLGA Nanofibers

During the kinetic experiment, bmMSC growth was evident in the PLGA nanofibers. Still, microscopic evidence could not be generated until SEM was employed (Figure 6). In PLGA nanofibers (Figure 6A,B), it was observed that there was a significant difference between the surface without cells (Figure 6A) where PLGA nanofibers are visible, and the surface with cells (Figure 6B) where PLGA nanofibers are covered by bmMSC growth. The distinct bmMSC growth morphologies observed between alginate microcarriers (Figure 6E,F) and SoloHill collagen-coated microcarriers (Figure 6C,D) during inverted microscope documentation were maintained. In alginate microcarriers, the alteration of classic MSC morphology observed in Figure 1 was confirmed with SEM (Figure 6E,F), in contrast to the enlarged fibroblastoid morphology observed in bmMSC grown in SoloHill collagen-coated microcarriers (Figure 6C,D).

### 3.7. Biocompatibility Test

Aqueous and hydrophobic extract from collagen coated-PLGA nanofibers and alginate microcarriers were obtained after an incubation at 37 °C for 72 h according to ISO 10993 [14], this is the best condition to evaluate the aqueous and hydrophobic molecules released by a medical device directly applied to the patient without any thermal sterilization in the production process. Figure 7 shows the obtained viability results for the tissues exposed to each treatment.

In reconstructed epidermis, tissues exposed to extracts from both materials (Figure 8C,D) more closely resemble the one exposed to negative control (Figure 8B) as in three tissues we can distinguish both stratum corneum and granulosum with positive staining of hematoxylin in the nucleus of living cells that form the stratum granulosum as opposed to tissue exposed to positive control (Figure 8A) where there is no sign of living cells stained with hematoxylin and also the substance has destroyed the stratum granulosum.

Additionally, in non-keratinized cornea, tissues exposed to extracts from both materials (Figure 8G,H) more closely resemble more the one exposed to negative control (Figure 8F) as in three tissues we can distinguish the characteristic 5–6 cell thick stratified squamous epithelium of non-keratinized cornea epithelium as opposed to tissue exposed to positive control (Figure 8E) with discontinuous separation zones inside the tissue and reparative atypia reflected in bigger cells with the formation of nucleoli.

## 4. Discussion

### 4.1. Alginate Microcarriers and Polylactic-Co-Glycolic Acid (PLGA) Nanofibers Production

In our study, PLGA and alginate were selected as the base molecules for the developed biomaterials due to the proven edibility and biocompatibility of these substances. On one hand, PLGA is an FDA-approved polymer for food and drug delivery usage, demonstrating biocompatibility by producing safe, non-toxic degradation products. Moreover, PLGA has been tested as an edible coating for fruits, enhancing their shelf life owing to its antimicrobial properties [21]. On the other hand, alginate has been explored as an edible material and finds diverse applications in the food industry, serving as a packaging material, encapsulation agent [22], thickener, emulsifier, and stabilizer for food [23]. The biocompatibility and edibility of PLGA and alginate make them suitable materials for growing animal cells, with potential applications in food, tissue engineering, cell therapy, and pharmaceutical applications. The size difference between alginate microcarriers and PLGA nanofibers is related to the used production technique [24]: in PLGA electrospraying, the jet from the Taylor cone is stabilized permitting elongation and the consequent formation of nanofibers. In contrast, electrospraying provokes destabilization of the jet and hence the formation of fine and homogenous droplets that result in the alginate microcarriers.

### 4.2. Disinfected and Collagen-Coated Alginate Microcarriers and PLGA Nanofibers Production

As alginate microcarrier production was performed in non-controlled microbiological conditions, a disinfection process was required. A 70% ethanol solution was used to effectively remove microorganisms while maintaining physical and chemical properties. This technique demonstrated efficiency and usefulness in laboratory-scale experiments. However, radiation can be employed as a large-scale disinfection method due to its efficiency and cost-effectiveness [25].

Phenol red, a widely used pH indicator for animal cell culture, was used in the MSC NutriStem XF media because substances during the coating and disinfection process could potentially affect the pH neutrality of the biomaterials. This pH neutrality test was useful in determining the suitability of PLGA nanofibers and alginate microcarriers for cell culture. Following the disinfection and collagen-coating treatments, the sterility and pH neutrality of the biomaterials were verified. Following a 48 h sterility test incubation, the cell culture media retained a red and clear appearance, indicating the absence of contamination, in contrast to the yellowish and turbid appearance observed when contamination occurred, accompanied by an acidic pH. This type of sterility testing is essential for all components that are intended for therapeutic use. Instead of employing the 14-day incubation period outlined in the pharmacopeial sterility test methodology, 48 h cell culture media incubation was utilized. The incorporation of a more nutritive media and a shorter incubation period fulfills the requirements for sterility testing of new-generation biologic products, facilitating a faster and more extensive detection of potential microbial contaminants.

Optic microscopic comparison of collagen-coated alginate microcarriers and Sartorius Solohill collagen-coated microcarriers in Figure 1E,F illustrates the difference of size between the two materials: 400 μm diameter of alginate microcarriers versus the 125–212 μm diameter of Sartorius Solohill collagen-coated microcarriers [26].

Besides the differences in size, an important distinction was observed in the cell morphology of bmMSCs growing in collagen-coated alginate microcarriers versus Sartorius Solohill collagen-coated microcarriers. It was noted that in alginate microcarriers, bmMSCs grew in a rounded morphology, contrasting with the enlarged and fibroblastoid-like morphology found in the Solohill collagen-coated microcarriers. The spindle-shaped morphology characteristic of MSCs was observed in the Solohill collagen-coated microcarriers. In the case of alginate microcarriers, the same rounded growth has been reported previously in cells encapsulated inside alginate beads and after cartilage differentiation in this material [27].

### 4.3. Alginate Microcarriers and PLGA Nanofibers Physicochemical and Structural Characterization

The variation in mass-loss temperatures could be related to chemical interactions between collagen and the base materials (alginate and PLGA). Coating the microcarriers with collagen can provide a protective barrier that improves thermal stability. Likewise, collagen could act as an insulating layer that protects the alginate from thermal degradation and allows the formation of new bonds, such as hydrogen bridges, dipole-dipole interactions, or other intermolecular forces [28]. As for PLGA, the addition of collagen could interfere with the PLGA nanofibers’ structure, affecting their thermal stability. The interactions between collagen and PLGA could weaken the polymeric matrix, resulting in lower thermal resistance. Thus, the decrease in thermal stability of collagen-coated PLGA nanofibers could be due to a combination of adverse molecular interactions, changes in morphology and structure, and the influence of thermal degradation of collagen at lower temperatures.

The temperature at which 50% mass is lost has been used to compare the effect of the material composition in the thermogravimetric properties [29]. Interestingly, collagen coating decreases the 50% mass-loss temperature for PLGA nanofibers from 400 to 300 °C. Such behavior is coherent to the results reported by Jose et al. [30], who produced electrospun PLGA/collagen blends for bone regeneration where the increase of collagen concentration in the coating causes the PLGA blend to resemble more the collagen TGA profile closely. In contrast, the 50% mass-loss temperature increased from 365 to 498 °C in alginate microcarriers. Previously, Hou et al. [31] found the same peaks and curve behavior in alginate microcarriers produced with different gold nanostar coating, varying only in the percentage of mass loss at each temperature. Although the addition of collagen may alter the thermal properties of PLGA nanofibers and alginate microcarriers, TGA indicates that both systems exhibit high stability within physiological temperature ranges. Therefore, the cell adhesion and TGA profile support the effectiveness of collagen coating for alginate microcarriers and PLGA nanofibers as growth surfaces for MSCs.

Figure 2C shows the infrared spectrum of dried samples of collagen-free and collagen-coated alginate microcarriers. The characteristic bands of alginate are evident, with peaks at 3344 and 2925 cm^−1^ corresponding to the stretching and weak aliphatic vibrations of the -OH and C-H groups, respectively. The band defined at 1040 cm^−1^ is associated with the vibration elongation of the C-O groups, while the vibrations at 1606 and 1406 cm^−1^ are linked to the antisymmetric and symmetric stretching of the COO- groups. Despite the low collagen concentration, vibrations associated with the protein are noticeable at 1548 to 1700 cm^−1^, primarily overlapping vibrations of the C=O groups present in both alginate and collagen [32].

Interestingly, certain regions of the spectrum are susceptible to changes after mixing alginate with collagen. An increase in transmittance is observed in the vibrations of the -OH groups after adding collagen, rising from 71% to 83%. Similarly, a significant shift is noted in this band, from 3367 to 3257 cm^−1^. Therefore, the increased transmittance and band shift after the addition of collagen may indicate alterations in the solvation of these groups, and hydrogen bonding interactions between the -OH groups of alginate and collagen structures. The IR profile similarities (Figure 2) between collagen-free and collagen-coated alginate microcarriers correspond to characteristic absorption peaks at 1616 and 1418 cm^−1^ for sodium alginate compounds [33]. This phenomenon is also evident in the bands at 1634 cm^−1^, which shifted to 1592 cm^−1^; its transmittance also changed from 75% to 72%. The persistence of such bands implies that the basic structure of alginate is maintained, but interactions with collagen could influence the relative intensity of these bands. Consequently, changes observed in the different bands suggest a potential molecular reorganization induced by the interaction between alginate and collagen. This interaction could be a primary factor contributing to the higher thermal stability observed in TGA (Figure 2B) in collagen-coated alginate microcarriers, as stronger interactions require higher energy for breakage. Finally, the collagen integration to this material affected the peak centered at 2900 cm^−1^, which can be attributed to the increase of L-proline as this amino acid has an absorption peak between 2745–3155 cm^−1^ [34] and constitutes about 10% of total amino acid composition of collagen [35].

### 4.4. Antibiotic and Serum-Free Mesenchymal Stem Cell Primary Culture and Flow Cytometry Characterization

The standard practice in mammalian cell culture is to use fetal bovine serum (FBS) as the main source of growth factors. The substitution of FBS is critical for therapeutic and food applications due to the price related to scarcity, zoonotic concerns, prevention of viral and prion contamination, and lot-to-lot reproducibility and industrial process standardization [36].

For meat production applications, FBS must be substituted due to sustainability and cruelty-free considerations. Previously, the use of serum-free media for porcine-induced pluripotent stem cells was reported for agricultural and therapeutic applications [37].

CD73, CD90, and CD105 were expressed in more than 89% of the cultivated bmMSCs. This immunophenotype coincided with CD90 expression in porcine MSCs [38], CD105 expression in bone marrow MSCs [39], and CD73 expression in porcine adipose and peripheral blood MSCs [40]. However, a lack of expression of CD34, CD45, and HLA-DR observed in the immunophenotype is consistent with the absence of hematopoietic markers [41] and the antigen-presenting receptor HLA-DR [42] expected for MSCs. The immunophenotype was determined by using a Stemflow Human MSC Analysis Kit, demonstrating cross-reactivity with porcine bmMSC surface markers. This cross-reactivity, not previously tested between porcine and human MSC surface markers, was supported by the kit’s prior cross-reactivity test conducted between human and mouse MSCs. These results endorse the use of Stemflow Human MSC Analysis Kits for characterization activities of porcine MSCs applications.

Adherent growth and the reported immunophenotype from cultured bmMSCs fulfil requirements of the International Society for Cellular Therapy (ISCT) for MSC characterization [43]. Additionally, Mycoplasma detection was performed to the bmMSC master bank as this microorganism is difficult to detect and undesirable for any of their applications [44]. As reported in Figure 3C, the evaluated bmMSCs were negative for Mycoplasma as filamentous fluorescence was not observed in the cytoplasm in contrast to the positive control.

The MSC source was bone marrow, obtainable from any location, minimizing the risk of microbiological contamination. This sample characteristic allows for antibiotic-free MSC culture, addressing health concerns for both food and therapeutic applications. The use of serum and antibiotic-free culture conserved the typical spindle-shaped morphology of bmMSCs as observed in Figure 3B.

### 4.5. Alginate Microcarriers and PLGA Nanofiber Growth Kinetics

Collagen has been extensively documented as a ligand that promotes cell adhesion in various scaffolds and materials [45]. Specifically, collagen has demonstrated robust adhesion to mesenchymal stem cells [46]. The significance of collagen coating has been previously reported for biomaterials as poly-L-lactic acid (PLLA) nanosheets [47]. Additionally, the absence of growth observed in collagen-free biomaterials (Figure 4) underscores the criticality of collagen coating in both PLGA nanofiber and alginate microcarriers.

Collagen coating has been proven critical for bmMSC adhesion as the collagen-free PLGA nanofibers and alginate microcarriers did not show any growth (Figure 4). The functionalization of biomaterials has been proven as necessary for enhanced MSC adhesion through the activation of FAK/Src, MAPK, PI3K/Akt, Wnt/B-catenin, and YAP/TAZ [48].

Collagen has been previously reported as a coating that increases cell viability and adhesion in biomaterials different from the experimental biomaterials [49]. The interaction of MSCs with collagen has been reported to increase the expression of genes associated with the production of extracellular matrix and adhesion proteins [50].

For these reasons, collagen coating has the potential to improve MSC proliferation and adhesion in different types of biomaterials as it has previously demonstrated to have interesting properties for therapeutic applications such as biomaterial bioresorbality [48] and co-coating with BMP-2 for osteogenic differentiation [51].

It can be concluded that the bmMSCs grow at the same rate and efficiency on both the nanofiber and cell culture plastic plates. These kinetic results support bmMSC growth adhered to PLGA nanofibers and will be confirmed by the SEM results (Figure 6). Furthermore, PLGA nanofibers offer several advantages for tissue engineering, including high solubility [52], high porosity, high specific surface areas [53], the potential for the material to form a nanocomposite serving as a sensor, actuator, and ion transporter [54], as well as the construction of three-dimensional microenvironments with high cell viability over extended periods for medical applications. Other forms of PLGA materials have been used for in situ applications like PLGA particles added to dendritic cells to enhance vaccine delivery [55]. Other electrosprayed biomaterials have reported therapeutic uses of MSC grown in nanofibers for skin wound healing, cardiac repair, urinary bladder regeneration, repair and regeneration of dental pulp, dentin, and periodontal tissue. Therefore, collagen-coated PLGA nanofibers developed during this study possess characteristics and therapeutic potential that necessitate further evaluation in preclinical models to assess their specific properties for tissue engineering.

As anticipated, a statistically greater percentage of relative growth efficiency was observed with both SoloHill collagen-coated microcarriers and collagen-coated alginate microcarriers compared to regular 12-well cell culture (Table 1). This is a consequence of optimizing the surface–volume ratio by using spherical microcarriers, contrasting with bidimensional surface MSC growth [56]. Finally, for the evaluation of the total 96 h incubation for all the treatments, Tukey test grouping for the final kinetic results of all treatments was performed (Table 1), and relative growth efficiency was calculated relative to 12-well cell culture surface (Table 2).

As indicated in Table 2, bmMSCs increased their number by almost five times after a 96 h incubation on the control surface (12-well cell culture). This result is higher than the reported 48 h doubling time in rat bmMSCs in FBS-supplemented media [57] and 54–65-h doubling time in porcine adipose MSCs [58]. This difference between FBS-supplemented and serum-free media has not been previously reported for porcine MSCs, as previous experiments demonstrated lower cell proliferation while maintaining the potential for differentiation [58].

The experimentally obtained cell population increases of 5.1 and 6.81 for SoloHill collagen-coated microcarriers and collagen-coated alginate microcarriers, respectively (Table 2), are close to the 5.47-fold expansion reported by Yan et al. [59] with human umbilical cord MSCs using serum-free medium and a porous microcarrier system (3D TableTrix). In Figure 4 and Appendix A Table A1, bmMSC grown in alginate microcarriers exhibit greater growth efficiency than commercial Solohill collagen-coated microcarriers cultivated in the same conditions.

This improvement in cell population from collagen-coated alginate microcarriers to commercially available microcarriers provides experimental support to evaluate the material on a larger scale to measure the effect on cell population fold. It has been widely reported that cell yield decreases with the increase in scale [60] as pH, dissolved oxygen, and shear stress significantly change in larger MSC production.

As shown in Figure 5, after reaching the maximum number of cells in each evaluated material after 96 h incubation, the number and viability of bmMSCs is maintained for 22 days. This condition is critical because cells can be maintained for a 3-week term, avoiding the increase of passages that cause replicative exhaustion, alteration in the differentiation potential, and progressive telomere length shortening [61].

The support of the bmMSC growth for 22 days from developed collagen-coated PLGA nanofibers and alginate microcarriers and the tested culture conditions also are compatible with the chondrogenic, osteogenic, and adipogenic differentiation protocols [62]. This condition, in addition to the biocompatibility properties tested in this article, supports the use of the bmMSCs grown in the developed materials for direct in situ applications.

### 4.6. Morphology of Alginate Microcarriers and PLGA Nanofibers

In the kinetic evaluation of bmMSC growth for alginate microcarriers, a considerable increase in cell growth was observed (see Appendix A, Table A1 and Table A2), but this did not align with the microscopic images of the microcarriers. Both properties suggest that bmMSCs grew inside the collagen-coated microcarriers. To confirm this, the alginate microcarriers were cut with scalpels to evaluate the adherence of bmMSCs inside. Figure 6G,H revealed the presence of bmMSCs with their traditional fibroblastoid morphology inside the alginate microcarriers. This microcarrier property, referred to as porosity, has been previously reported in other edible and biocompatible cell attachment materials such as colloidal lignin particles [63] and cellulose nanofibril alginate hydrogels [64].

### 4.7. Biocompatibility Test

As shown in Figure 7A,B the evaluated extracts and direct applications of collagen coated-alginate microcarriers and PLGA nanofibers did not cause any significant effect on the viability of non-keratinized cornea and reconstructed epidermis. As negative and positive controls as well as vehicles comply with the expected results and validated methods for skin [19] and ocular irritation [20] were used, it can be concluded that collagen-coated alginate microcarriers and PLGA nanofibers are ocular and skin non-irritants.

These results were confirmed by histological hematoxylin and eosin staining of exposed non-keratinized cornea and reconstructed epidermis; as shown in Figure 8.

As far as we know, our study represents one of the first experiments with porous edible microcarriers, showing a statistically significant increase in cell growth compared to classic bidimensional growth surfaces and commercially available microcarriers. Additionally, the experiments were performed with bmMSCs grown in antibiotic- and serum-free media, facilitating the integration of the reported protocol into current practices and requirements within the biotech and food industries. Unlike other large-scale MSC production techniques such as SoloHill collagen-coated microcarriers and based on the cytocompatibility and biocompatibility of the microcarriers developed in this study, alginate microcarriers with attached MSCs can be evaluated for direct therapeutic applications to promote nerve [65] and bone regeneration, and cartilage engineering and repair [66].

The edibility of alginate enables the evaluation of developed microcarriers with attached MSCs as fundamental blocks for cultivated meat production (Song et al., 2022), as spherical microcarriers are recognized as the best options for scaling up MSC production in the context of cultivated meat [9]. Therefore, the next step for this application will be the evaluation of the differentiation of MSCs grown in the alginate microcarriers into myoblasts, complying with food regulations throughout the entire process. This meat production edible microcarrier application has been thoroughly evaluated by Yen et al. [67]; in contrast to our research, their study uses fetal bovine serum-supplemented medium and non-porous microcarriers.

## 5. Conclusions

bmMSCs used for this article comply with the ISCT minimal criteria for multipotent mesenchymal stromal cells for immunophenotype and growth characteristics; additionally, were tested for the absence of bacteria, fungus and Mycoplasma.

Collagen-coated alginate microcarriers and PLGA nanofibers have been proven as feasible biomaterials to support their use in the MSC production process for their multiple applications in the therapeutic and food industry. Both materials were proven skin and ocular non-irritant following ISO-10993 guidelines and validated protocols to assess their biocompatibility.

The collagen integration caused changes in the thermogravimetric and infrared profile of the alginate microcarriers and PLGA nanofibers. This collagen coat is critical for the adherence of the bmMSCs to the biomaterials. No statistically significant differences were detected in the bmMSC growth in PLGA nanofibers and 12-well cell culture plates. In contrast to plastic surfaces, PLGA nanofibers have flexibility, porosity, and high specific surface area that support the possibility of forming nanocomposites and the formation of tridimensional microenvironments for extended-period MSC culture and in situ applications in clinical and therapeutic applications.

During the optic and scanning electronic microscopy visualization of bmMSCs growing in the collagen-coated alginate microcarriers, the observed cell morphology was round as opposed to the expected spindle-shaped form. These observations, in addition to the increase in bmMSC growth as compared to other biomaterials, were hints that indicated that bmMSCs grew inside the collagen-coated alginate microcarriers. This theory was confirmed by the SEM documentation of cut collagen-coated alginate microcarriers. Inside the microcarriers the bmMSCs grew with the expected fibroblastoid morphology. This condition explains the increase of bmMSC growth with collagen-coated alginate microcarriers and supports that the developed biomaterial can be classified in the porous microcarrier category.

The relevance of the high bmMSC production yield inside an edible and biocompatible material is based on the ease to integrate the alginate microcarriers with cells into the final product: tissue, organs, and cultivated meat. This characteristic will increase efficiency and reduce costs as the MSCs will not require a trypsinization or other detachment method.

The potential integration of alginate microcarriers for MSC production for cell therapy, tissue engineering, and cultivated meat will support economically feasible applications and a better optimization of the surface–volume ratio for the bioreactors required to produce these cells in large scales. Further research is required to evaluate the suitability of the developed collagen-coated PLGA nanofibers and alginate microcarriers system to grow MSCs in larger scales. Additionally, further design and testing of specific in situ therapeutic and food applications based in the developed biomaterials should be explored.

## Figures and Tables

**Figure 1 biomolecules-15-00345-f001:**
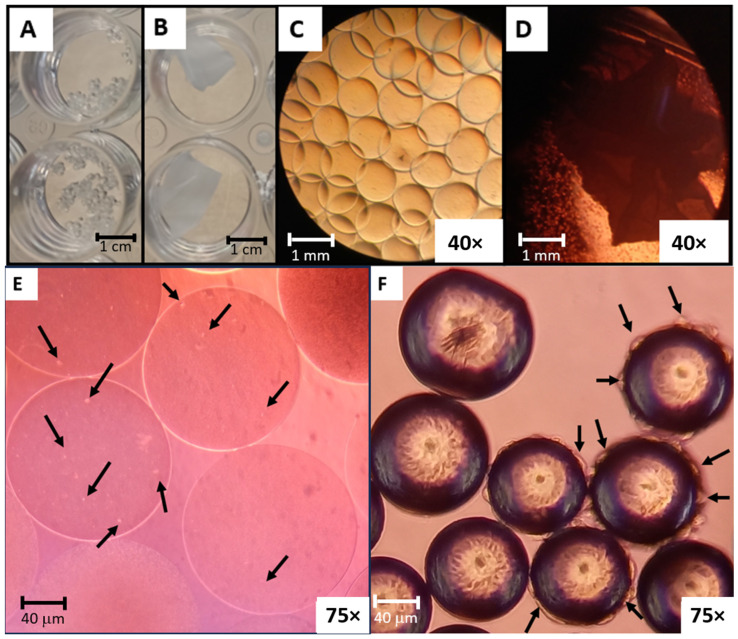
Morphology at different scales of the alginate microcarriers and PLGA nanofibers: (**A**) macroscopic morphology of alginate microcarriers and (**B**) PLGA nanofibers; microscopic morphology of (**C**) alginate microcarriers (40× objective) and (**D**) PLGA nanofibers. bmMSCs adherence confirmation test (40× objective): (**E**) cells attached to alginate microcarriers are signaled with black arrows (75× objective); (**F**) cells attached to Solohill collagen-coated microcarriers (75× objective) are highlighted by black arrows.

**Figure 2 biomolecules-15-00345-f002:**
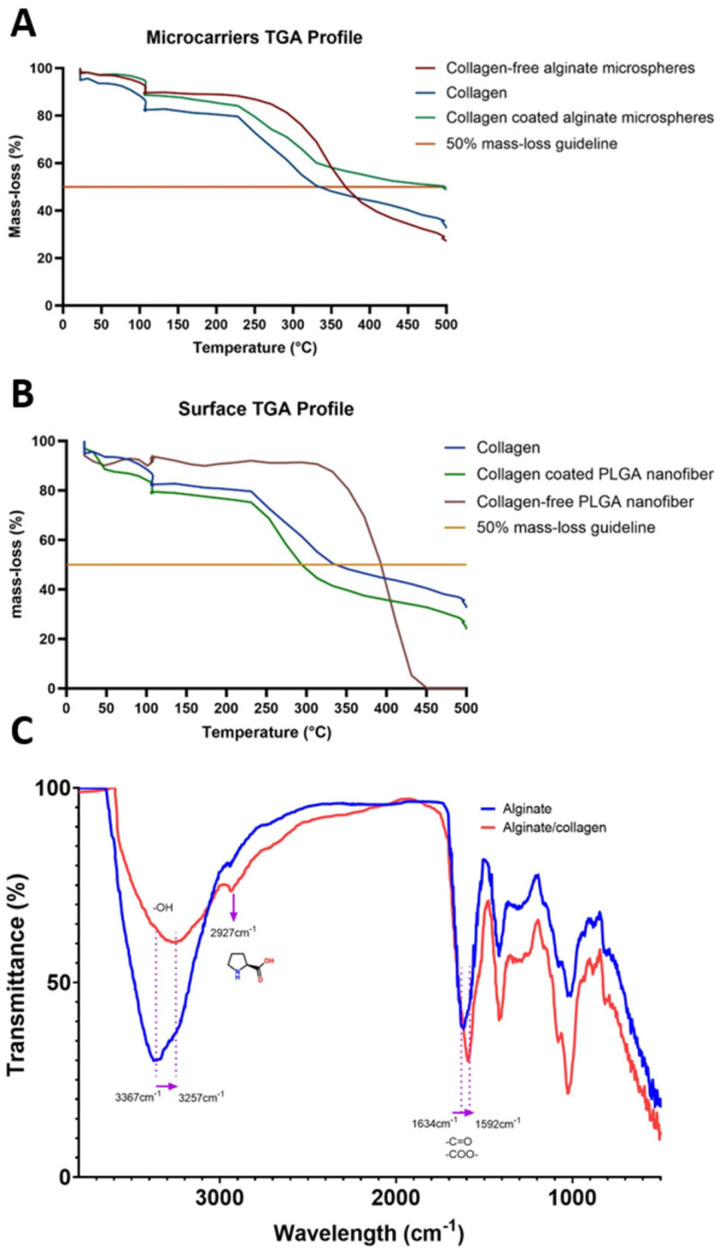
Thermogravimetric analysis curve of microcarriers (**A**) and surface (**B**)**.** In this figure, collagen-free alginate microcarriers, PLGA nanofiber, collagen and collagen-coated alginate microcarriers were included. (**C**) Comparative Fourier Transform Infrared (FT-IR) spectra from collagen-free and collagen-coated alginate microcarriers. Collagen-free alginate microcarrier FT-IR spectrum is shown in blue and collagen-coated alginate spectrum in red.

**Figure 3 biomolecules-15-00345-f003:**
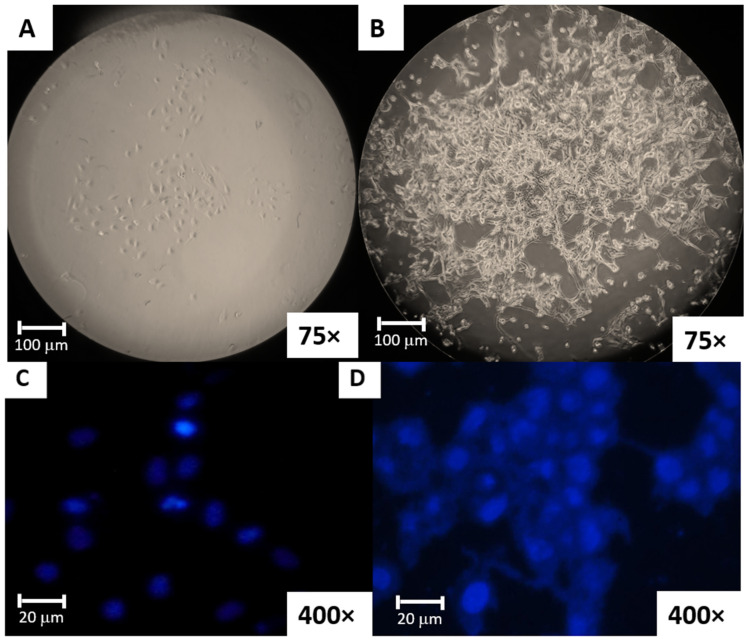
Porcine bone marrow bmMSC microscopic morphology. (**A**) First passage bone marrow bmMSCs (75× objective). (**B**) Passage 10 bmMSCs (75× objective). Indirect Hoechst staining results: (**C**) Master bank porcine bmMSCs negative for Mycoplasma (400× objective). (**D**) Mycoplasma contaminated bmMSCs (400× objective).

**Figure 4 biomolecules-15-00345-f004:**
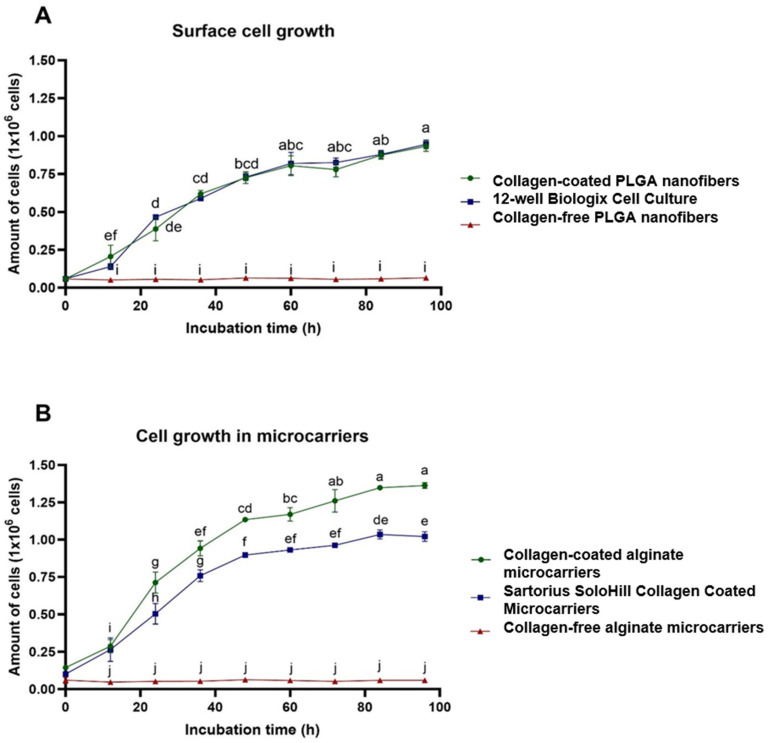
Comparative growth kinetics for microcarriers and surfaces evaluated. In both cases, time was plotted in the *x*-axis and the average of total cells ± SD in the *y*-axis. (**A**) Microcarrier kinetic. (**B**) Surface kinetic.

**Figure 5 biomolecules-15-00345-f005:**
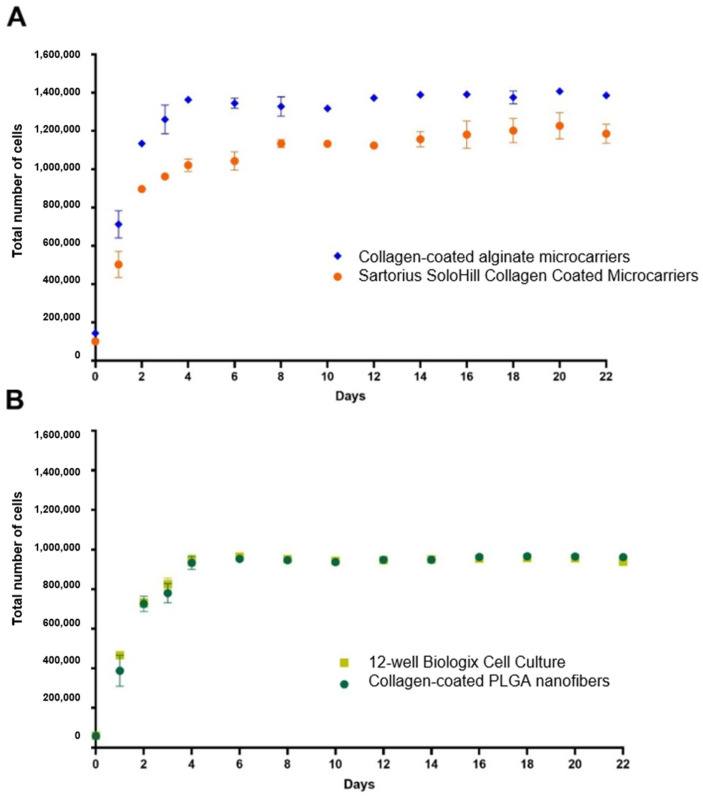
Comparative 22-day growth kinetics for microcarriers and surfaces evaluated. In both cases, time was plotted in the *x*-axis and the average of ±standard deviation in the *y*-axis. (**A**) Microcarrier kinetic. (**B**) Surface kinetic.

**Figure 6 biomolecules-15-00345-f006:**
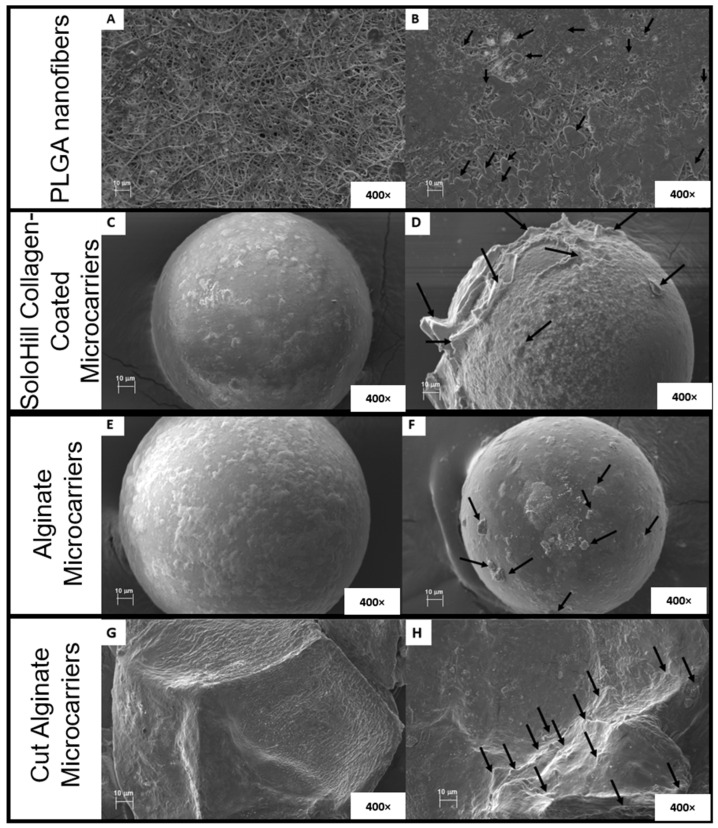
Morphology of evaluated systems with and without cells. (**A**) PLGA nanofiber without cells; (**B**) PLGA nanofiber with cells; (**C**) SoloHill collagen-coated microcarrier without cells; (**D**) SoloHill collagen-coated microcarrier with cells; (**E**) alginate microcarriers without cells; (**F**) alginate microcarriers with cells; (**G**) cut alginate microcarriers without cells; (**H**) cut alginate microcarriers with cells. All cells were documented with 400× objective and cells are highlighted with black arrows.

**Figure 7 biomolecules-15-00345-f007:**
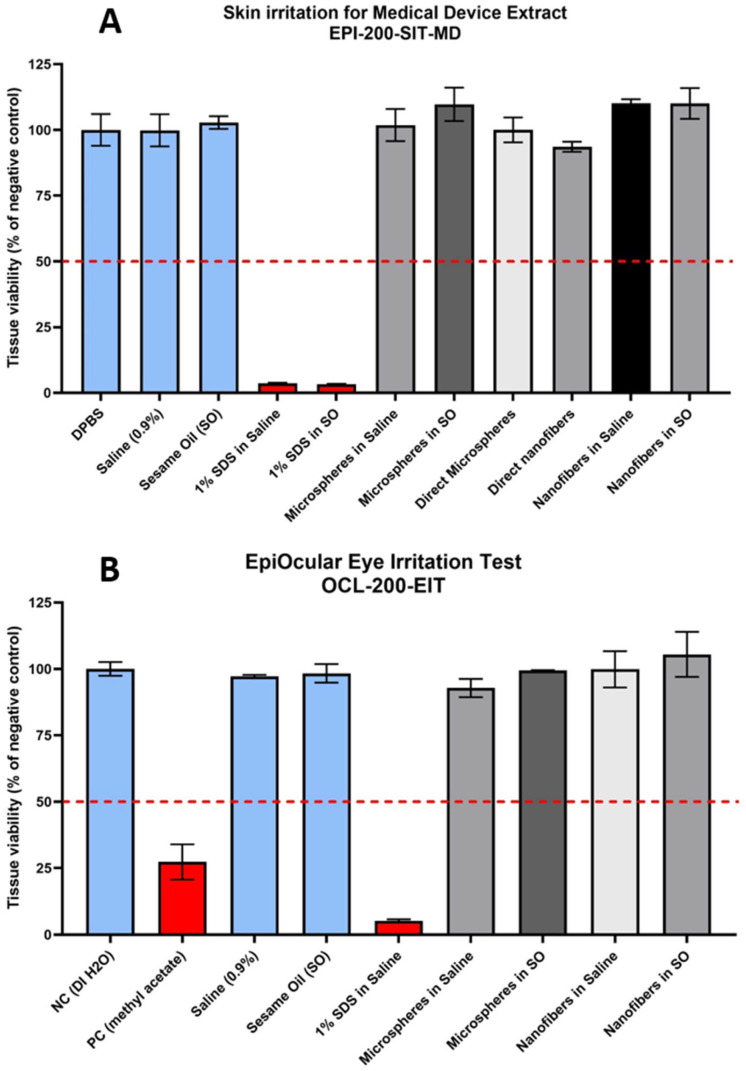
(**A**) Skin irritation results. The 50% viability threshold for irritant and non-irritant is marked in red. Evaluated samples were: negative control (PBS), vehicles (0.9% saline and sesame oil), positive controls (1% SDS in 0.9% saline and 1% SDS in sesame oil), collagen-coated alginate microcarriers (0.9% saline extract, sesame oil extract and direct application), collagen-coated PLGA nanofibers (direct application, 0.9% saline extract and sesame oil extract). (**B**) Ocular irritation results. The 60% viability threshold for irritants and non-irritants is marked in red. Evaluated samples were: negative control (distilled water), positive control (methyl acetate), vehicles (0.9% saline and sesame oil), positive control (1% SDS in 0.9% saline), collagen-coated alginate microcarriers (0.9% saline extract and sesame oil extract), collagen coated-PLGA nanofibers (0.9% saline extract and sesame oil extract).

**Figure 8 biomolecules-15-00345-f008:**
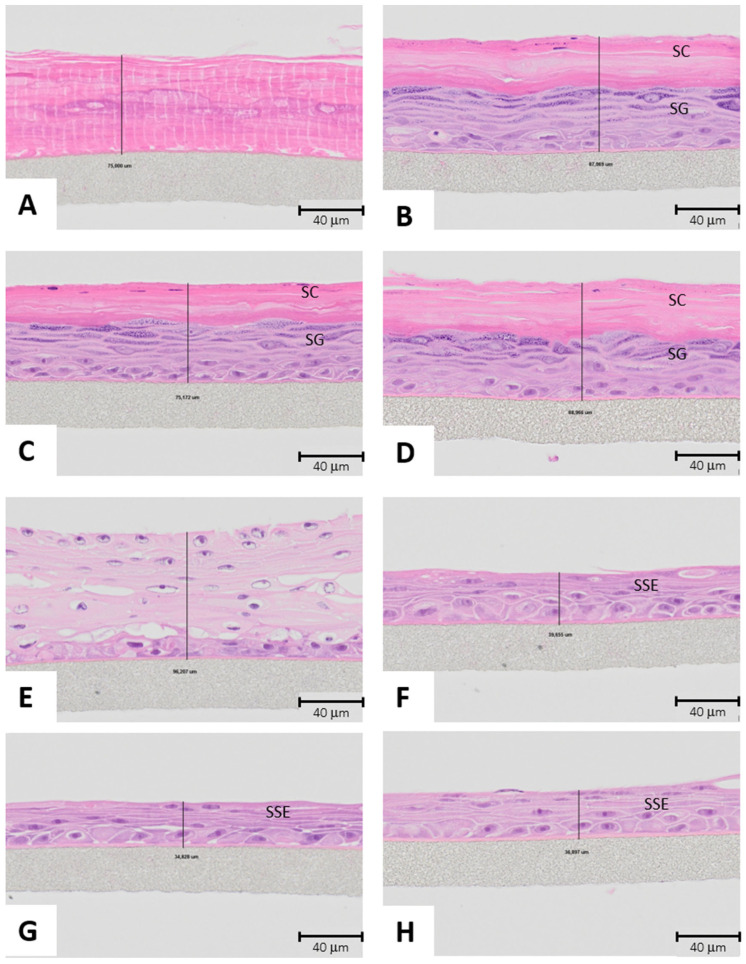
Reconstructed epidermis hematoxylin and eosin stain. (**A**) Positive control (1% SDS in Saline), (**B**) negative control (PBS), (**C**) qqueous extract from collagen coated-alginate microcarriers and (**D**) aqueous extract from collagen coated-PLGA nanofibers. Non-keratinized cornea hematoxylin and eosin stain at 40× exposed to (**E**) positive control (methyl acetate), (**F**) negative control (distilled water), (**G**) aqueous extract from collagen coated-alginate microcarriers, and (**H**) aqueous extract from collagen coated-PLGA nanofibers. All samples are visualized with an 40× objective. SC: Stratum corneum, SG: Stratum granulosum and SSE: Stratified Squamous Epithelium.

**Table 1 biomolecules-15-00345-t001:** Tukey test grouping after 96 h incubation for each treatment.

Treatment	Total Cells (Mean ± Standard Deviation)
Collagen-coated alginate microcarriers	1,362,500 ± 0.0194 ^A^*
SoloHill collagen-coated microcarriers	1,021,167 ± 0.0324 ^B^*
12-well cell culture plate	945,167 ± 0.0287 ^C^*
Collagen-coated PLGA nanofibers	932,633 ± 0.0330 ^C^*
Collagen-free PLGA nanofibers	65,233 ± 0.0029 ^D^*
Collagen-free alginate microcarriers	58,733 ± 0.0027 ^D^*

* Letters indicate the statistical groups obtained by Tukey test.

**Table 2 biomolecules-15-00345-t002:** Total cell and relative growth efficiency for 96 h incubation for the treatments evaluated.

Treatment	Total Cells at Initial Time	Total Cells at 96-h Incubation	96-h Cell Population Increase	% Relative Growth Efficiency
Collagen-coated alginate microcarriers	200,000	1,362,500	6.81	144.15
SoloHill collagen-coated microcarriers	200,000	1,021,167	5.1	108.04
Collagen-coated PLGA nanofibers	200,000	932,633	4.7	98.67
12-well cell culture	200,000	945,167	4.7	100
Collagen-free PLGA nanofibers	200,000	65,233	0.3	6.9
Collagen-free alginate microcarriers	200,000	58,733	0.3	6.21

## Data Availability

The original contributions presented in this study are included in the article. Further inquiries can be directed to the corresponding author.

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
