# Peer review of "Evaluation of Biocompatible Materials for Enhanced Mesenchymal Stem Cell Expansion: Collagen-Coated Alginate Microcarriers and PLGA Nanofibers"

_biomolecules, 2025, doi:10.3390/biom15030345_

Round 1
Reviewer 1 Report
Comments and Suggestions for Authors
This article aims to evaluate the efficacy of electrosprayed alginate microcarriers and electrospun PLGA nanofibers, along with their collagen-coated iterations, as highly efficient proliferation surfaces for bone marrow-derived MSCs. The manuscript is well-written, requiring few revisions, and it is highly relevant to the field of regenerative medicine. Please correct capitalization, grammatical, as well as abbreviation errors, and address the comments below:
Introduction - Add discussion on different sources of MSCs and how different MSCs (e.g., adipose, bone marrow, and umbilical cord-derived) could yield different therapeutic effects. Add references: https://doi.org/10.1038/s41536-019-0083-6, https://doi.org/10.1002/stem.2575
Throughout the paper, specify the MSC type being cited from a study.
For the MSCs used in this study, label them as bone-marrow derived throughout the paper.
Materials - Specify the molecular weight of polymers used, especially alginate and PLGA.
Results and discussion -3.1 Alginate microcarriers and Polylactic-co-glycolic acid (PLGA) nanofibers production. Add discussion on why the collagen-coated alginate is smaller in size.
For all figures - Define all abbreviations used in the caption, and include scale bars even for photographs and light microscopy images.
Results and discussion - 4.5 Alginate microcarriers and PLGA nanofiber growth kinetics. Add discussion on possible mechanisms by which collagen can improve MSC expansion. Addition of natural polymers can improve MSC adhesion, proliferation, and secretion through the activation of multiple signaling pathways, such as FAK/Src, MAPK, PI3K/Akt, Wnt/β-catenin, and YAP/TAZ. Add references: https://doi.org/10.31083/j.fbl2906228
For all tables - Define all abbreviations used in the footnote.
Results and discussion - 4.5 Alginate microcarriers and PLGA nanofiber growth kinetics. Add discussion on how collagen has the potential to improve the expansion of MSCs not only on alginate and PLGA, but also other polymers that are sprayed or spun, with already proven benefits for various medical applications. Add references: https://doi.org/10.1002/adhm.202300960, https://doi.org/10.1016/j.colsurfb.2020.111040
Figure 8. Label or annotate the relevant layers instead of just showing total thickness.
Author Response
Dear Editors and Reviewers:
Thank you for reviewing our manuscript entitled "Evaluation of Biocompatible Materials for Enhanced Mesenchymal Stem Cell Expansion: Collagen-coated alginate microcarriers and PLGA nanofibers” (ID: biomolecules-3471400).
We have responded in detail to all the comments of revisors, and We appreciate the valuable and helpful comments provided by the reviewers. We believe that addressing these comments has improved the quality of the manuscript substantially.
We have included as a revised version the final version with audit trail and additional images for the text.
The response to each observation is the following:
Reviewer #1:
- Introduction - Add discussion on different sources of MSCs and how different MSCs (e.g., adipose, bone marrow, and umbilical cord-derived) could yield different therapeutic effects. Add references: https://doi.org/10.1038/s41536-019-0083-6, https://doi.org/10.1002/stem.2575
Thank you for your comment. We have added the discussion of different sources of MSC and their different therapeutic potential in Lines 56-68. Additionally, we have added the two references suggested in this space.
- Throughout the paper, specify the MSC type being cited from a study.
Thank you for this observation. We conducted a thorough revision to specify bone marrow-derived mesenchymal stem cells (bmMSCs) where applicable while retaining the term MSCs for general references to this cell type. All modifications have been highlighted for your review.
- For the MSCs used in this study, label them as bone-marrow derived throughout the paper.
We greatly appreciate this suggestion. To refer to the MSCs used in this research we used the term bmMSCs (bone marrow-derived mesenchymal stem cells) throughout the article.
- Materials - Specify the molecular weight of polymers used, especially alginate and PLGA.
Thank you very much for this comment. The molecular weight and a more accurate description of the used alginate and PLGA has been added to the Materials section in Lines 141-143.
- Results and discussion -3.1 Alginate microcarriers and Polylactic-co-glycolic acid (PLGA) nanofibers production. Add discussion on why the collagen-coated alginate is smaller in size.
We are thankful for your observation. We added an explanation of the size difference between alginate microspheres and PLGA nanofibers in Lines 359-364.
- For all figures - Define all abbreviations used in the caption and include scale bars even for photographs and light microscopy images.
We kindly recognize this suggestion. We added scale bars for Figure 1, 3 and 8 and all abbreviations used in the caption were defined in the footnote and integrated in the Abbreviation section.
- Results and discussion - 4.5 Alginate microcarriers and PLGA nanofiber growth kinetics. Add discussion on possible mechanisms by which collagen can improve MSC expansion. Addition of natural polymers can improve MSC adhesion, proliferation, and secretion through the activation of multiple signaling pathways, such as FAK/Src, MAPK, PI3K/Akt, Wnt/β-catenin, and YAP/TAZ. Add references: https://doi.org/10.31083/j.fbl2906228
Thank you for your comment. We have added the mechanisms of the improvement of cell adhesion with collagen coating and the recommended bibliography in Lines 440-453.
- For all tables - Define all abbreviations used in the footnote.
Thanks for this observation. All abbreviations used in Tables were defined in the footnote and integrated in the Abbreviation section.
- Results and discussion - 4.5 Alginate microcarriers and PLGA nanofiber growth kinetics. Add discussion on how collagen has the potential to improve the expansion of MSCs not only on alginate and PLGA, but also other polymers that are sprayed or spun, with already proven benefits for various medical applications. Add references: https://doi.org/10.1002/adhm.202300960, https://doi.org/10.1016/j.colsurfb.2020.111040
We greatly appreciate this suggestion. We have added the mentioned collagen property to the Discussion section in Lines 440-453 along with the recommended references.
- Figure 8. Label or annotate the relevant layers instead of just showing total thickness.
Thank you very much for this comment. We label the relevant layers in Figure 8.
The specific description and localization of the whole corrections are enlisted in the previous ten reviewer observations responses. Additionally, a wide ortographic and redaction review has been performed.
We would like to thank you once again for your time and effort in reviewing our manuscript and hope that our manuscript will now be considered suitable for publication in Biomolecules Journal.
Thanks for your consideration,
Kind regards.
Dr. R. Alejandra Chávez Santoscoy

Reviewer 2 Report
Comments and Suggestions for Authors
This study investigated collagen-coated alginate microcarriers and PLGS nanofibers for scalable MSC production. These offer efficient, biocompatible alternatives to traditional growth surfaces in regenerative medicine and cultivated meat manufacturing. Results showed physicochemical characterization, in vitro tests, mycoplasma tests, and in situ applications, which clarified the novelty of this investigation. I have some comments in this manuscript:
- Introduction: The overview section is long and lacks research objectives. It needs to be more concise.
- Materials and methods:
- Consistent notation of the sources of chemical materials used in the study is required.
- The experimental methods should be briefly summarized, and supplementary content can be included in the supplementary data.
- Results:
- Figure 1: please add the scale bar for figures C, D, E, and F.
- Figure 3: please add the scale bar for figures A, B, C, and D.
Author Response
Dear Editors and Reviewers:
Thank you for reviewing our manuscript entitled "Evaluation of Biocompatible Materials for Enhanced Mesenchymal Stem Cell Expansion: Collagen-coated alginate microcarriers and PLGA nanofibers” (ID: biomolecules-3471400).
We have responded in detail to all the comments of revisors, and We appreciate the valuable and helpful comments provided by the reviewers. We believe that addressing these comments has improved the quality of the manuscript substantially.
We have included as a revised version the final version with audit trail and additional images for the text.
The response to each observation is the following:
Reviewer #2:
- Introduction: The overview section is long and lacks research objectives. It needs to be more concise.
We greatly appreciate this suggestion. We rewrote Lines 122-135 to clarify the research objectives. We deeply reviewed the introduction and we found that all the information was relevant to understand the article content, moreover Reviewer #1 requested to add information and references about MSCs source effect in therapeutic applications.
- Materials and methods:
Consistent notation of the sources of chemical materials used in the study is required.
Thanks for this observation. We added relevant missing information from chemical materials with emphasis in the used polymers in Lines 141-143.
- The experimental methods should be briefly summarized, and supplementary content can be included in the supplementary data.
We kindly recognize this suggestion. We summarized the methodology, eliminating redundancy and useless details in Lines 177-297.
- Results:
Figure 1: please add the scale bar for figures C, D, E, and F.
Figure 3: please add the scale bar for figures A, B, C, and D.
Thank you very much for this comment. We added the scale bars to Figure 1 and 3.
The specific description and localization of the whole corrections are enlisted in the previous ten reviewer observations responses. Additionally, a wide ortographic and redaction review has been performed.
We would like to thank you once again for your time and effort in reviewing our manuscript and hope that our manuscript will now be considered suitable for publication in Biomolecules Journal.
Thanks for your consideration,
Kind regards.
Dr. R. Alejandra Chávez Santoscoy

Round 2
Reviewer 1 Report
Comments and Suggestions for Authors
All the issues have been resolved.